# Effects of Homogeneous and Heterogeneous Crowding on Translational Diffusion of Rigid Bovine Serum Albumin and Disordered Alfa-Casein

**DOI:** 10.3390/ijms241311148

**Published:** 2023-07-06

**Authors:** Aleksandra M. Kusova, Ilnaz T. Rakipov, Yuriy F. Zuev

**Affiliations:** 1Kazan Institute of Biochemistry and Biophysics, FRC Kazan Scientific Center, Russian Academy of Sciences, Lobachevsky Str. 2/31, Kazan 420111, Russia; alexakusova@mail.ru; 2Institute of Chemistry, Kazan Federal University, Kremlevskaya Str. 18, Kazan 420008, Russia; ilnaz0805@gmail.com

**Keywords:** proteins, homogeneous and heterogeneous crowding, diffusion, PFG NMR

## Abstract

Intracellular environment includes proteins, sugars, and nucleic acids interacting in restricted media. In the cytoplasm, the excluded volume effect takes up to 40% of the volume available for occupation by macromolecules. In this work, we tested several approaches modeling crowded solutions for protein diffusion. We experimentally showed how the protein diffusion deviates from conventional Brownian motion in artificial conditions modeling the alteration of medium viscosity and rigid spatial obstacles. The studied tracer proteins were globular bovine serum albumin and intrinsically disordered α-casein. Using the pulsed field gradient NMR, we investigated the translational diffusion of protein probes of different structures in homogeneous (glycerol) and heterogeneous (PEG 300/PEG 6000/PEG 40,000) solutions as a function of crowder concentration. Our results showed fundamentally different effects of homogeneous and heterogeneous crowded environments on protein self-diffusion. In addition, the applied “tracer on lattice” model showed that smaller crowding obstacles (PEG 300 and PEG 6000) create a dense net of restrictions noticeably hindering diffusing protein probes, whereas the large-sized PEG 40,000 creates a “less restricted” environment for the diffusive motion of protein molecules.

## 1. Introduction

The real protein environment in living systems significantly influences their conformational, functional, and dynamic properties [1,2]. The in vivo water compartments are saturated by a large number of macromolecules with concentrations from 80 to 400 g/L, occupying up to 40% of the water volume and creating macromolecular crowding [3,4]. Molecular crowding leads to an increase in excluded volume effects, a rise of viscosity, and the growth of specific and non-specific intermolecular interactions [5,6,7,8,9,10,11]. Significant changes in protein structure, diffusion transfer, and functioning were revealed under crowding conditions [12,13,14,15]. Crowding is expected to be essentially important for intrinsically disordered proteins (IDPs), due to their conformational malleability, tendency to aggregation, and openness for the targets binding [16,17].

Historically, protein biochemistry and biophysics were aimed toward the study of structurally rigid proteins when their well-defined 3D conformation determines protein functioning according to the structure−function paradigm. However, in the last years, evidence has shown that many proteins or their functional regions manage without intrinsic rigidity, operating by the highly dynamic mobility of their structure [18]. The classical example of such IDP is the chaperone-like activity of milk caseins, when the fuzzy casein structure blockades the unfolding of targeted proteins and prevents their afunctional aggregation [19,20]. Diffusion is the fundamental physical phenomenon determining functional properties of rigid and IDP proteins via their interaction with environment and specific recognition in living systems, for which it is necessary to know how to model them and to take into account.

Among many research approaches to estimate the effects of macromolecular crowding on protein structure, stability, and behavior, the analysis of protein translational diffusion is of great significance, since the diffusive behavior of tracer (proteins in our case) can serve as the signature of protein interaction with its environment. The diffusion of proteins in complex environments has been studied many times in different time and length scales in various environments in diverse model systems [21,22,23,24,25,26,27,28,29]. In addition, the difference in protein molecular size and shape, and the presence of specific and non-specific intermolecular interactions of proteins were taken into account [30]. Many attempts to characterize proteins’ non-specific intermolecular interactions are known which represent proteins mainly in terms of rigid particles with corresponding excluded volumes and restricted diffusive motion [6,31,32,33]. The tested interactions between particles involved the hydrodynamic term in addition to the excluded volume and particle–particle collisions. The obtained experimental data on protein diffusion under crowding conditions deviates significantly from most of phenomenological predictions [34,35,36]. Thus, we were stimulated to look for new solutions for the diffusive motion of biomacromolecules in a complex interior.

In homogeneous dilute medium, the translational diffusion of rigid particle *D_0_* is described usually by the Stokes–Einstein equation for particle Brownian motion:(1)$$D_{0}=\frac{k_{B}T}{6\pi \eta R_{h}},$$
where *k_B_* is the Boltzmann constant, *T* is the absolute temperature, *R* is the particle hydrodynamic radius, and *η* is the medium dynamic viscosity [37]. The Stokes–Einstein law works rather well for many dilute systems including protein solutions [22,24]. However, in concentrated/crowded solutions the protein self-diffusion was shown to deviate strongly from the Stokes–Einstein predictions [23,38,39]. The deviation can be caused by the non-Newtonian behavior of solutions or the increase of excluded volume effect in the presence of crowding agents [40]. In addition, the concentration of protein systems under study affects the stabilization/destabilization of proteins and resulting protein–protein and protein–crowder interactions [41,42,43]. The physical consequence of macromolecular crowding declares itself mainly in hard-core and weak non-covalent interactions [25]. The hard-core interactions represent a steric effect arising from the impenetrable nature of atoms and molecules, which reduces the available free volume for their diffusive motion [44,45]. Non-covalent interactions are usually caused by the uneven and patched distribution of interfacial charge and hydrophobic regions over the protein surface, which determine the balance of different interactions between protein molecules and crowding agents [46,47]. 

One of the most common experimental procedures to study different aspects of macromolecular crowding in vitro involves synthetic and natural crowders [5,48,49,50]. Their effect is usually considered as the excluded volume of uncharged hard spheres [51,52]. However, besides the effect of the excluded volume, recent publications also indicated the influence of weak non-covalent interactions of high-molecular crowding agents on the diffusion of tracer biomacromolecules [43,53]. Several studies have shown that the use of inert polymers (PEGs, dextrans, and Ficolls) to mimic the in vitro cell interior is ambiguous [44,54,55,56], although there is the ability to use these synthetic polymers [17,57,58,59,60].

In the present work, we compared the influence of a crowding environment on self-diffusion of rigid globular protein, bovine serum albumin (BSA), and intrinsically disordered protein (IDP) α-casein, with the one-order hydrodynamic radius (3.2 nm for BSA and from 3.5–8 nm for α-casein due to its possible little oligomerization), which differed basically in their structural organization [61,62]. Functional properties of α-casein are controlled by a balance of hydrogen bonding, hydrophobic interactions, and electrostatic repulsion that depends on the surrounding proteins [63]. Because of the lack of its rigid tertiary conformation, α-casein can reply very rapidly to environmental changes that allow the protein to interact with multiple target molecules. The determination and explanation of protein diffusion is of great importance for determining the crowding medium influence on protein diffusion in model and living systems. Herein, with the help of pulsed-field gradient (PFG) NMR, we compared the effects of concentrated homogeneous and heterogeneous environments created by glycerol and PEGs, respectively, on protein translational diffusion [26,64]. We observed self-diffusion coefficients of rigid globular protein BSA and IDP α-casein in crowding environments and compared the protein’s behavior in crowding environments with the Stokes–Einstein predictions. The crowded environment was modeled using synthetic biocompatible polyethylene glycol (PEG) polymers over a wide range of molecular weights (300–40,000 Da). The PEGs create excluded volume effects and dehydrate biopolymers at high concentrations via water deficiency and hydration effects [45]. PEGs are chemically inert for proteins allowing them to observe the crowding effect without a weak interaction between the crowding agent and the test protein [65]. Different crowding agents (with more than three orders in molecular weight) have shown effects on protein translational behavior when changing their concentration from a dilute to semi-diluted polymer regime, simulating various crowding environments found in a heterogeneous biological environment. The PEGs clustering and aggregates formation is directly dependent upon the H bonds, their entropy, and the hydrophobic interaction balance [66]. The net-like structure based on polymer aggregates depending on concentration and temperature, as well as the chain length and chain termination, were shown for PEGs with various molecular weights [67,68,69]. The use of structured and disordered proteins makes it possible to compare the effect of a concentrated macromolecular medium on protein–protein interactions depending on the structural organization of proteins. The obtained results indicated the discrepancy between the experiment and predictions for protein diffusion in the PEG water solutions. To describe protein self-diffusion behavior in PEG systems, we applied a phenomenological generalization of the “tracer on lattice” model [70,71]. This model was originally invented for the diffusion of tracers in a regular lattice. To modify this model for protein diffusion in solutions we introduced some fitting parameters to characterize the deviation of the spatial distribution of obstacles from that in the case of a regular lattice. The fitting algorithm, considering a three-component system (protein, crowder, and solvent), was based on the relationship between observed protein self-diffusion coefficients and phenomenological parameters, which contains information about the spatial distribution of obstacles created by crowder molecules.

We determined the fundamental differences in the effect of low- and high-molecular crowding agents on protein self-diffusion. It was found that the largest deviation from the Stokes–Einstein predictions was caused by the PEG with smaller molecular mass. The “tracer on lattice” model showed that the low-sized PEGs (300 and 6000 Da) operate protein diffusion by the creation of a regular network, while the deviation from this model was observed in the case of large-sized PEG 40,000. It was also shown that heterogeneous (PEG 300/PEG 6000/PEG 40,000) crowding agents used in this work do not have fundamental differences in their effect on rigid BSA and unstructured α-casein.

### Theory

The Stokes–Einstein relation (Equation (1)), being truly valid only at infinite dilution of non-interacting spherical particles, may be used to describe the hydrodynamic behavior of chemical species from its diffusion coefficient *D*_0_. For concentrated solutions, the relation between particle hydrodynamic size and its self-diffusion coefficient *D* is more complicated. The diffusing species often interact with the environment via the hydrodynamic and non-covalent intermolecular contacts causing a strong deviation of the self-diffusion coefficient from the Stokes–Einstein predictions [29,32,51,72]. On one hand, the experimental protein diffusion coefficient can display an increase in solution viscosity. We checked this effect using viscous glycerol as the solvent component. 

The theoretical description of diffusion in inhomogeneous media is an extremely complex problem. In contrast to the case of a homogeneous environment, macromolecular diffusion in heterogeneous media is characterized by the diffusion coefficient as well as the form of the diffusion equation, which depends on types of inhomogeneity. Many approaches to the theoretical description of various types of inhomogeneity have been proposed. For instance, “anomalous” diffusion, when the root-mean-square displacement of a particle <∆*r*^2^> over time *t*, changes according to the law:<∆*r*^2^> = 2*dD^a^t^a^,*(2)
where *d* is the space dimension and *D* is the self-diffusion coefficient. 

The case *a* = 1 corresponds to normal diffusion in a homogeneous medium and 0 < *a* < 1 and *a* > 1 correspond to cases of subdiffusion and superdiffusion, respectively. Subdiffusion is used, for example, for the phenomenological description of diffusion processes in porous media [73,74,75,76]. This approach does not provide an understanding of the details of the process on a molecular level.

There is an approach that considers the diffusion coefficient in systems with obstacles (hindered diffusion) based on hydrodynamic and thermodynamic relations [77]. It has been proposed for the diffusion of spherical colloidal particles in rigid polymer gels. For the diffusion coefficient *D*, the authors [77] obtained a complex expression, which goes into the well-known expression in the limit of volume fraction of obstacles *φ_f_* → 0 [78]:(3)$$D_{0}=\frac{k_{B}T}{6\pi \eta R_{h}}\left(1+1.45\varphi +O\left(\varphi^{2}\right)\right),$$
where *φ* is the volume fraction of spherical particles of radius *R_h_*. 

In the limit of infinite dilution *φ* → 0 and finite *φ_f_*, their expression gives another limiting case of the diffusion coefficient in gels *D = D*_0_
*F* where *F* is a steric hindrance or tortuosity factor [79,80]. This approach imposes strict restrictions on the shape of diffusing molecules.

There is a wide range of models from the theory of diffusion in dispersed media. For example, Maxwell–Garnett [81] and Fricke [82] models are used for the case of spherical obstacles; the Bruggemann model [83] is applicable for a mixture of spheres or circular cylinders of observed particles and obstacles (both phases are equal). All these approaches are limited by the low concentration of obstacle molecules.

For a theoretical description of our experimental results for heterogeneous medium with hard restrictions, we applied the “tracer on lattice” model [70,71]. It was developed for the diffusion of a tracer restricted by a regular isotropic lattice. In our case, the tracer is a protein molecule, and the obstacles are formed by crowder molecules of PEGs. We consider the obstacle molecules as electrically neutral so their hindrance to the diffusing tracer is solely due to the exclusion volume effect. In the case of mobile obstacles, the tracer diffusion coefficient is a function of both concentration of obstacles (*φ*) and their relative jump rate *γ* [70,71]. A theoretical expression for *D*(*φ*, *γ*) was obtained by Tahir-Kheli, Van Beijeren, and Kutner [71,84]. The diffusion coefficient *D* can be expressed in terms of the correlation factor *f*(*φ*, *γ*):(4)D=(1−φ)f(φ,γ)
where: $$f(\varphi ,\gamma )=\frac{{\left\{{\left[(1-\gamma )(1-\varphi )f_{0}+c\right]}^{2}+4\gamma (1-\varphi )f_{0}\right\}}^{1/2}-\left[(1-\varphi )(1-\gamma )f_{0}+\varphi \right]}{2\gamma (1-\gamma )f_{0}},\;\gamma =\frac{\Gamma_{p}}{\Gamma_{q}}\Rightarrow \gamma =\frac{D_{p}}{D_{q}},$$
where *c* is the probability of the lattice site to be occupied by a tracer molecule, *f*_0_ = (1 − *α*)/[1 + *α*(2*γ* − 1)] and *α* is the asymmetry parameter, which depends on the lattice: for a square lattice *α* = 1 − 2/*π* [85]. The role of this parameter is in taking into account the deviation of the diffusion process in the solution from that on a regular lattice. *Γ_p_* is the rate of the tracer jumps, *Γ_q_* is the rate of the obstacle jumps, *D_p_* is the diffusion coefficient of protein (tracer), and *D_q_* is the diffusion coefficient of the obstacle molecule.

## 2. Results and Discussion

### 2.1. Impact of Size-Dependent Crowding Agents on Translational Diffusion of Rigid BSA

We started our findings with an analysis of the viscous effect, applying the glycerol–water solvent to model the role of medium viscosity. Figure 1 depicts experimental values of the BSA self-diffusion coefficient in water-glycerol solutions in comparison with ones calculated according to the Stokes–Einstein equation. Values of BSA hydrodynamic radius [86] and viscosity of water-glycerol solutions [87,88,89] were required for calculations. One can see that experimental and calculated values of *D* are close to 40% glycerol content. The coincidence of curves in this concentration range indicates the agreement with conditions of particle diffusion in a continuous viscous medium and affects BSA diffusion only due to increased viscosity of the protein solution [28]. The further discrepancy of experiment data from calculations may be caused by the limitation of the PFG NMR method due to the insufficient strength of the magnetic field gradient providing the high error of *D* measurements. Moreover, very intense crowding agent signals not only obscure nearby signals but also deleteriously affect the S/N of the desired solute component thus resulting in sensitivity loss [90].

To study the influence of the high-molecular crowders size on the translational diffusion of proteins, we applied PEGs, which are commonly used to model highly crowded cellular environments [29,42,91,92,93]. Figure 2 shows the self-diffusion coefficient of BSA as a function of PEG mass fraction. It should be noted that in the case of PEGs, a more intense decrease of *D* of BSA is observed compared to glycerol solutions, which may be associated with the two-dimensional effect of a complicated environment on protein diffusion. Figure 2 shows the significant sharp influence of PEG 40,000 on the BSA self-diffusion coefficient compared to PEG 300 and PEG 6000. 

First of all, we estimated the viscosity effect in the PEG water solutions on BSA diffusion. To calculate the *D* using the Stokes–Einstein relation (Equation (1)), we need to use the viscosities of the PEG water medium. The values of dynamic viscosity of PEG 300/PEG 6000/PEG 40,000 water solutions, obtained by viscometry, are presented in Figure 3. Significant increases in the viscosity of solutions of higher-molecular PEGs were detected. 

We estimated the *D* of BSA using experimental data on the viscosity of PEG solutions (Figure 3). One can see that the experimental and predicted Stokes–Einstein equation values of *D* for BSA do not coincide for all PEG water solutions in the studied concentration range of crowding PEGs (Figure 4). It is known that crowding agents can influence the discrepancy of experimental data with the Stokes–Einstein predictions [28,29]. The results obtained indicate the different types of discrepancies between experimental and calculated values of the BSA self-diffusion coefficient for different PEGs. In an effort to understand the reason for such behavior, we applied the phenomenological “tracer on lattice” model [70] to fit BSA self-diffusion data.

The fitting of the experimental results presented in Figure 5 show that the “tracer on lattice” model (Equation (4)) fit well the obtained data on BSA self-diffusion coefficient in PEG 300 and PEG 6000 water solutions (Figure 5A,B). This theoretical approach allowed us to estimate protein behavior in the surrounding obstacles with the help of the contingent parameter “*a*” that takes into account the deviation of diffusion process in the crowding solution from one on a regular lattice. The asymmetry parameter *a* is useful to distinguish directed motion from Brownian diffusion, but not to resolve the structure of different types of obstacles. However, authors [85] showed that *a* → 0 corresponds to a free diffusion whereas *a* → 1 describes a fully restricted diffusion regime. In the case of *a*~0.1, it suggests trapping of the tracer molecule in a bounded region. One can see that the “tracer on lattice” model coincides well with the obtained experimental data of PEG 300 and PEG 6000 water solutions in the entire concentration range demonstrating the existence of a regular “lattice network” on the way of diffusing protein molecules. The apparent dense packing of PEG 300 and PEG 6000 relates to their smaller size (PEG 300: *R_h_* = 0.74 nm, PEG 6000: *R_h_* = 2.5 nm) compared to a protein molecule (*R_h_* = 3.2 nm). [94,95]. However, the “tracer on lattice” model does not describe adequately the influence of PEG 40,000 on BSA translational diffusion, possibly due to another mechanism of crowder influence on protein diffusion (Figure 5C). When the size of the PEG 40,000 crowder molecule (*R_h_* = 3.95 nm) exceeds the size of the protein molecule, the latter does not perceive restrictions in the form of a regular lattice, having free space between crowders (Figure 6). It has been shown that systems with small crowders are described by large surface areas creating ordered restrictions for diffusing probes [96]. As a result, the diffusion of protein molecules in such systems differs more strongly from Stokes–Einstein predictions compared with those with larger protein crowders. The crowder size should be viewed when describing various kinetic properties of biomacromolecules in a dense environment [96,97].

### 2.2. Translational Diffusion of Unstructural α-Casein in Crowded Solution

We carried out the same experiments with IDP α-casein to compare the behavior of proteins with various structural organizations in a crowded environment. Both α-casein and BSA were characterized by a sharper slowing-down of *D* in the PEG 40,000 water solution compared with PEG 300/6000. Similar dependences were observed for normalized *D* values of α-casein and BSA (Figure 7). 

With the help of the Stokes–Einstein relation, we estimated *D* of α-casein using experimental data on viscosity for PEG solutions (Figure 3). The experimental and theoretical dependences for α-casein self-diffusion coefficient as a function of PEG 300, PEG 6000, and PEG 40,000 wt. concentrations are presented in Figure 8. The experimental data and Stokes–Einstein predicted that α-casein *D* does not coincide for all PEG-water systems in the studied concentration range. Due to the obtained discrepancy between experimental and calculated values, we also applied the phenomenological “tracer on lattice” model [70] to explain these results.

Figure 9 shows the “tracer on lattice” model fitting (Equation (4)) of obtained data on α-casein self-diffusion in PEG 300/PEG 6000/PEG 40,000 water solutions. The “tracer on lattice” model coincides well with the obtained experimental *D* of α-casein in PEG 300 and PEG 6000 water solutions. In these systems, α-casein molecules are restricted in the close to regular lattice network. The same values of parameter “*a*” (deviation of the diffusion process in the solution from regular lattice) for α-casein and BSA in PEG 300/6000 water solutions were observed. Again, as with rigid BSA, big PEG 40,000 molecules cannot form the regular lattice for α-casein, which molecules are notably smaller than crowder. Consequently, the IDP α-casein and rigid structured BSA are sensing the created restrictions in a similar way. It should be noted that in our previous work, we have shown the fundamental difference in translational diffusion of structured rigid and disordered proteins in self-crowding environment [61,98]. We observed significant role of flexible domains of α-casein in the self-diffusion process due to a high friction between protein molecules leading to their inter-entanglement, whereas we did not detect significant effect of flexible fragments of α-casein on translational diffusion in inert crowding environment.

Because of their worm-shaped architecture, the structure of IDPs seems to be extremely sensitive to changes in their environment and could be responsive to the presence of crowding agents. Since different types of conformational behavior have been revealed for IDPs in crowding conditions, these proteins have been recently divided into three different groups: (partially) foldable (able to assume a partial fold in crowded milieu), non-foldable (insensitive to crowding conditions) and un-foldable (IDPs which undergo unfolding due to crowded environment) [17,99,100,101,102]. Previously it was shown that α-casein belongs to a non-foldable IDP [103]; thus, we proposed that the identical behavior of IDP α-casein as ordered BSA can be caused by a non-foldable α-casein state in a crowded milieu. Moreover, the known examples of using crowding agents with different IDPs have shown that proteins remain disordered in both the presence of exogenous crowding agents and the crowded cellular environment [17,102].

## 3. Materials and Methods

### 3.1. Chemicals

Glycerol (Sigma-Aldrich, Burlington, MA, USA) and polyethylene glycols (TatChemProduct, Kazan, Russia) of various molecular mass 300 Da (PEG 300), 6000 Da (PEG 6000), and 40,000 Da (PEG 40,000) were chosen as crowding agents. For experiments with crowding systems the glycerol/water and PEG/water solutions were prepared with agitation by magnetic stirrer. Water solutions were prepared using mQ-water (Direct-Q System, Millipore, Andover, MA, USA). The concentrated stoke solutions of lyophilized BSA (50 mg/mL) and α-casein (50 mg/mL) (Sigma-Aldrich, Burlington, MA, USA) were prepared in 100% D_2_O. Aliquots of 120 μL protein stock solution were poured into crowding agent–water (H_2_O) mixtures (480 μL) to attain total volume 600 μL and protein concentration 10 mg/mL. For all protein solutions, the pH was adjusted by microliter amounts of NaOH and HCl to a physiological value 7.4. Such a pH value was used to prevent aggregation of proteins [61,104]. All experiments were conducted at 298 K.

### 3.2. NMR-PFG Measurements

The protein self-diffusion coefficients were determined using NMR spectrometer (AVANCE-III, Bruker, Billerica, MA, USA) operating at 600.13 MHz equipped with a standard z-gradient inverse probe TXI (5 mm) capable of producing magnetic gradients up to a maximum strength 55.7 G cm^−1^. The ^1^H NMR spectra were obtained by nonselective (zggpw5) and selective (selpse) pulse sequences. Diffusion decays were obtained with the help of a modified Stejskal–Tanner pulse sequence (selgpse) to observe proton signals of proteins having low intensity compared to intense proton signals of crowding agents. The selgpse pulse sequence was applied to obtain selective ^1^H spectra with 128 scans, 4 dummy scans, a receiver gain of 32, an acquisition time of 2.7 s, and a relaxation time of 5 s. The length of the 180-degree shaped pulse was 2.8–4.0 ms. The excitations were performed for the δ region 0.42–2.23 ppm for intense signal of protein aliphatic protons. NMR data were processed with the help of TopSpin 3.6.1 software (Bruker Biospin Corporation, Billerica, MA, USA). In Figure 10, one can see the ^1^H spectra of α-casein in solution with PEG 6000 weight content of 9% obtained by nonselective “zggpw5” and “selpse” pulse sequences, respectively. Figure 10A shows that intense signals of the crowding agent protons prevent the observation protein component. At the same time, a selectively selected spectrum range (0.42–2.23 ppm) allows one to observe the region containing only protein signals (Figure 10B). 

The signal intensities of BSA and α-casein protons as a function of pulsed field gradient value are presented in Figure 11. The mono-exponential initial slopes of diffusive decays for BSA and α-casein were used to obtain *D* for all studied protein/crowding agent–water solutions [105]. 

### 3.3. Viscosity Determination

The dynamic viscosity of experimental solutions was measured at 298.15 K using the viscometer Lovis 2000 ME (Anton Paar, Graz, Austria). The apparatus was calibrated with water following the instructions and requirements of the manufacturer. The temperature was controlled by the built-in Peltier system to within ±0.005 K. The resolution of flow time measurements was 0.001 s. The viscosity reproducibility was within 0.5%. The uncertainty for viscosity determination for 1.59 mm capillary and steel ball was 0.007.

## 4. Conclusions

We investigated the translational diffusions of proteins in homogenous (glycerol solutions) and heterogeneous (PEGs solutions) media using the pulsed field gradient (PFG) NMR. The tracer probes used were globular bovine serum albumin and intrinsically disordered α-casein. The viscosity of PEG water solutions was obtained independently using the viscometer method.

Translational diffusion of spherically shaped BSA seems to obey the Stokes–Einstein model in homogenous glycerol solutions over the range of glycerol 40 mass percent. In the solution of low-weight PEG300, the translational diffusion of BSA and α-casein exhibits a large deviation from the Stokes–Einstein model for Brownian diffusion. The comparison of proteins’ behavior in heterogeneous crowded environments with the Stokes–Einstein predictions showed a deviation from Brownian motion depending on the structure of the diffusing species and surrounding media. The observed deviation from the Stokes–Einstein model might be attributed to excluded volume effects and intermolecular protein–protein and protein–crowder interactions. To fit the experimental data of self-diffusion coefficients, the “tracer on lattice” model was applied, where the fitting *a*-value was in taking into account the deviation of the diffusion process in the solution from that on a regular lattice. The applied “tracer on lattice” model showed that smaller crowding obstacles (PEG 300 and PEG 6000) created a dense net of restrictions noticeably hindering diffusing protein probes, whereas the large-sized PEG 40,000 created a “less restricted” environment for the diffusive motion of protein molecules.

Different IDPs, which show very distinct responses to crowded conditions, can be grouped into three categories: (partially) foldable, non-foldable, and unfoldable by macromolecular crowding. Based on our findings of similar self-diffusion coefficient dependences for unstructured α-casein as rigid ordered BSA on crowder content, we hypothesized that α-casein belongs to a non-foldable IDP. Fonin et al. reported [17] that only some IDPs fold in crowded conditions. It was proposed that the global structure of many unstructured proteins seems to be insensitive to the presence of crowding agents, since the anticipated overall contribution of macromolecular crowing to medium total free energy is not too high.

## Figures and Tables

**Figure 1 ijms-24-11148-f001:**
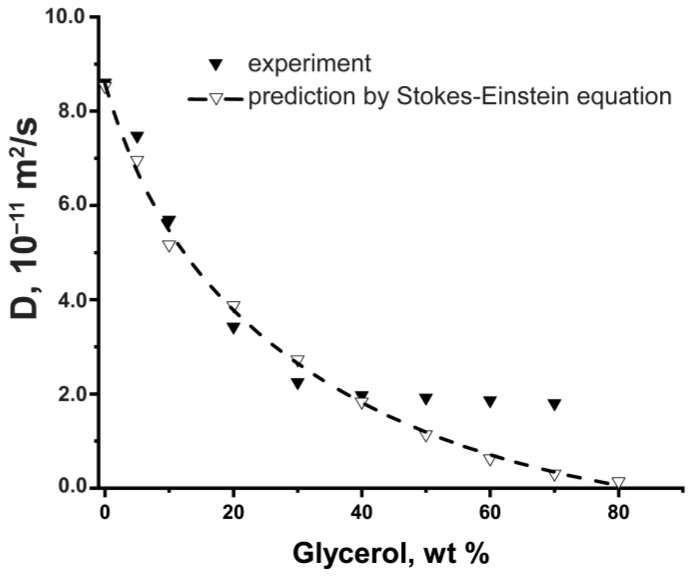
Experimental and predicted by Stokes–Einstein equation self-diffusion coefficient of BSA (10 mg/mL) in water–glycerol medium as a function of the glycerol mass percent.

**Figure 2 ijms-24-11148-f002:**
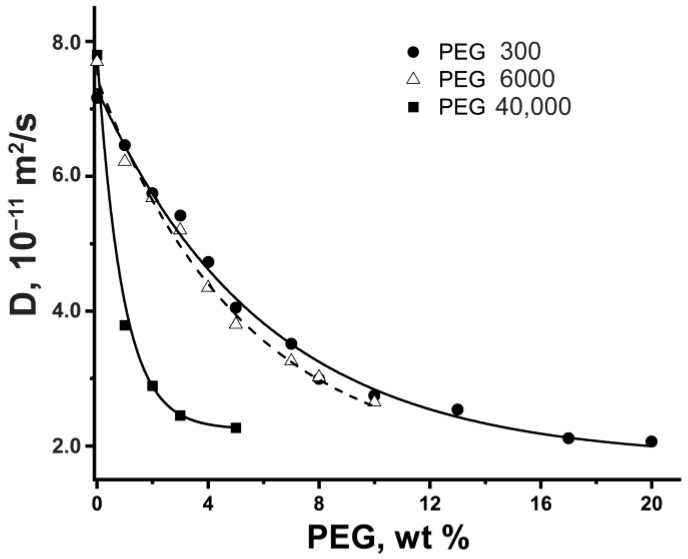
Experimental self-diffusion coefficients of BSA (10 mg/mL) as a function of mass percent of PEG 300 (circles), PEG 6000 (triangles), and PEG 40,000 (squares). Lines are guides for the eyes.

**Figure 3 ijms-24-11148-f003:**
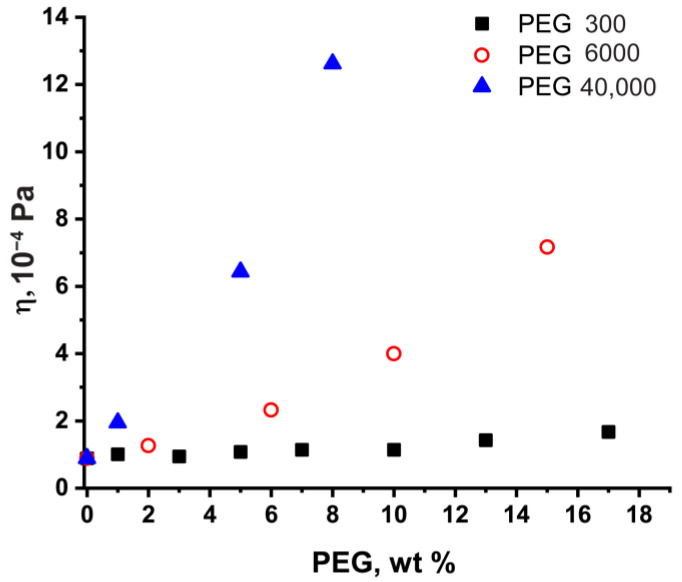
Dynamic viscosity of PEG water solutions at 298 K.

**Figure 4 ijms-24-11148-f004:**
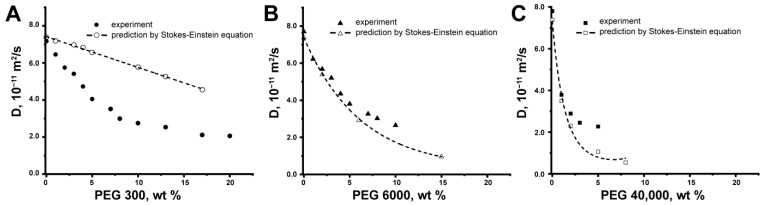
Experimental and predicted by Stokes–Einstein self-diffusion coefficients of BSA (10 mg/mL) as a function of mass percent of (**A**) PEG 300, (**B**) PEG 6000, and (**C**) PEG 40,000.

**Figure 5 ijms-24-11148-f005:**
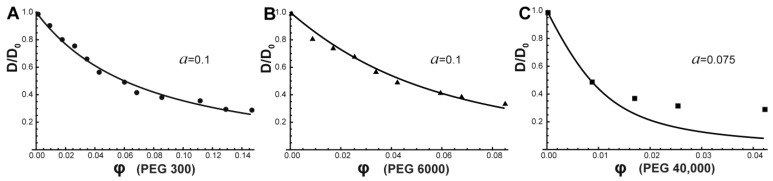
Normalized self-diffusion coefficient of BSA as a function of crowding-agent volume fraction for (**A**) PEG300, (**B**) PEG 6000, and (**C**) PEG 40,000. *D*_0_ represents BSA self-diffusion coefficient in pure water solution. Solid lines show the result of model fitting.

**Figure 6 ijms-24-11148-f006:**
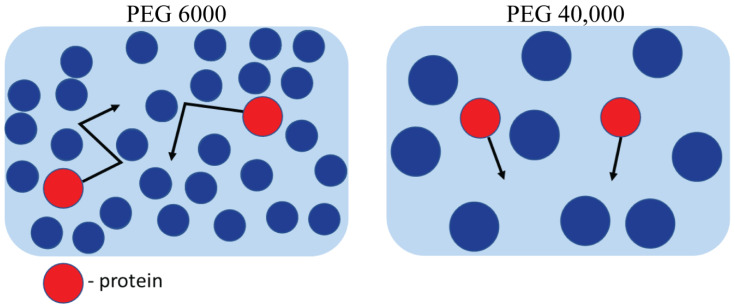
Simplified two-dimensional molecular composition of “tracer on lattice” model. Red particles represent protein and blue particles are crowder molecules of appropriate size. The more probable direction of protein motion is represented by arrows.

**Figure 7 ijms-24-11148-f007:**
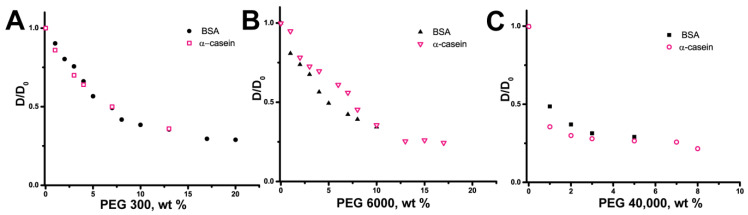
Normalized self-diffusion coefficient *D* of α-casein (10 mg/mL) and BSA (10 mg/mL) as a function of (**A**) PEG 300 (**B**) PEG 6000 (**C**) PEG 40,000 wt. concentration.

**Figure 8 ijms-24-11148-f008:**
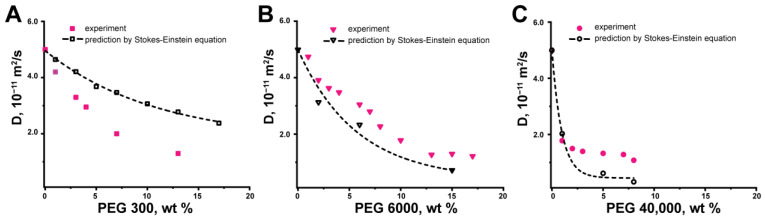
Experimental and predicted Stokes–Einstein self-diffusion coefficients of α-casein (10 mg/mL) as a function of mass percent of (**A**) PEG 300, (**B**) PEG 6000, and (**C**) PEG 40,000.

**Figure 9 ijms-24-11148-f009:**
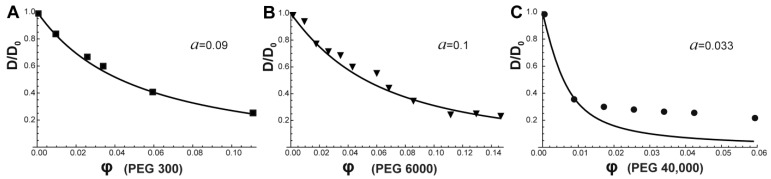
Normalized self-diffusion coefficients of α-casein as function of volume fraction of PEG 300 (**A**), PEG 6000 (**B**), and PEG 40,000 (**C**). *D*_0_ represents α-casein self-diffusion coefficient in pure water solution. Solid lines show result of model fitting.

**Figure 10 ijms-24-11148-f010:**
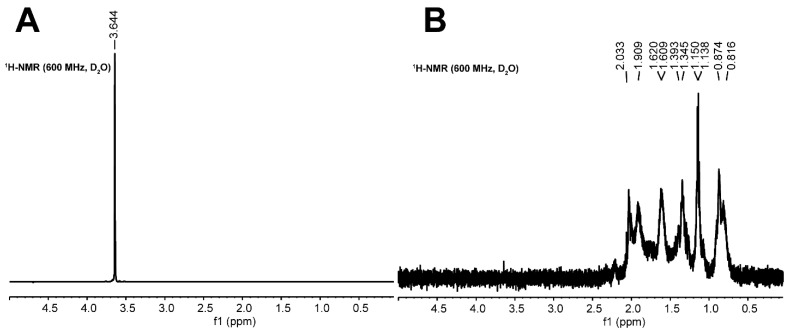
^1^H NMR spectrum (D_2_O, 600 MHz): δ_H_ 0.00–5.00 spectra of α-casein in solution containing 9 wt. % of PEG 6000 obtained by (**A**) nonselective (zggpw5) and (**B**) selective (selgpse) pulse sequences.

**Figure 11 ijms-24-11148-f011:**
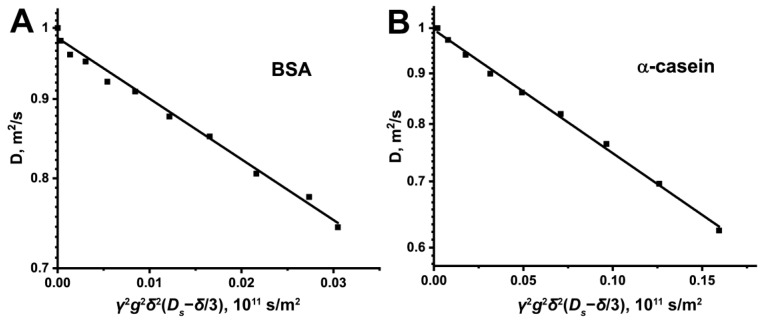
Initial part of diffusion decays (D_2_O, 600 MHz): (**A**) δ_H_ 0.45–2.65 (BSA) and (**B**) 0.53–2.52 (α-casein) of proton signal intensity for proteins in crowded solutions obtained by selective gradient excitation.

## Data Availability

The data in this study are available on reasonable request from the corresponding author.

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
