# Peer review of "Effects of Homogeneous and Heterogeneous Crowding on Translational Diffusion of Rigid Bovine Serum Albumin and Disordered Alfa-Casein"

_ijms, 2023, doi:10.3390/ijms241311148_

Round 1

Reviewer 1 Report

In the present manuscript, the authors tried to unravel the effects of homogeneous and heterogeneous, crowded environment of protein diffusion studies. The results obtained indicated the different type of discrepancy between experimental and calculated values of BSA self-diffusion coefficient for different PEGs. As an effort to understand the reason of such behavior they applied the phenomenological “tracer on lattice” model to fit BSA self-diffusion data. The “tracer on lattice” model fits well the obtained data on BSA self-diffusion coefficient in PEG 300 and PEG 6000 water solutions but not for PEG 40000. Both the IDP α-casein and rigid structured BSA are sensing the created restrictions in a similar way. The authors Revealed the system that something which contradicts Stokes-Einstein predictions. Likewise, there are interesting insights described by the authors which lead them to pave the way for many logical hypotheses based on the distinct responses of IDP’s to crowded conditions. They tried maximum possible usage of (PFG) NMR for their studies. The presentation and characterizations are very well done and shown in the figures very clearly. These studies may help future protein engineering studies like post translational modifications or binding studies and interactions of ligands to proteins etc.  Overall, the manuscript can be acceptable to publish in IJMS as such.

Author Response

Dear Reviewer!

Thank You for Your appreciation of our article.

Sincerely, 

Yuriy Zuev

Reviewer 2 Report

Kusova and colleagues proposed a paper entitled " Effects of homogeneous and heterogeneous crowded environment on protein diffusion studied by 1H PFG-NMR."

This paper presents pulsed-field gradient liquid-state NMR measurements of the hydrodynamic diffusion coefficients of two proteins, a globular protein, BSA, and a putatively intrinsically disordered protein (IDP), alpha-casein, as a function of increasing medium crowding. They increase the crowding of the medium by increasing the concentration of the Poly Ethylene Glycol (PEG) polymer.

The results are interesting: proteins no longer follow the Stokes-Einstein diffusion equation for very crowded media (not surprisingly). The authors show that an equation taking into account PEG diffusion as a diffusion barrier provides a better approximation of the behavior, except for high-mass PEGs.  The two proteins behave similarly, even though they have different overall structures, in principle. The biological significance of the article needs to be reviewed, as the results do not support the discussion on IDPs.

Unfortunately, the article is not very well structured, nor clear and logical, and significant changes are needed to make it acceptable for publication.

The following detailed notes are intended to help authors:

Title: In my opinion, the title is a little too general, since the authors only studied BSA and alpha-casein.

Page 2 and 3 Equations 1 and 2. Both equations define D as the diffusion constant, and in the text the authors refer to Do and Ds. Please clarify this point and add the correct subscripts to equations 1 and 2.

Page 2 lines 67-70. The use of the qualifier "soft" to describe the electrostatic interaction and the hydrophobic effect is a little clumsy. This qualifier is not necessary.

Page 2 Theory section. The authors describe hindered diffusion using the tracer on lattice model, which is correct. However, there are other models for hindered diffusion, such as cluster theory (see for example Zhang et al. 2023 ACSNano or Kamgar et al, J Mol Liquids, 2017). It would be interesting to discuss why these other models have not been used.

Page 4. The parameter c is not defined in equation.

Page 4 line 165. PFG limitation. What is the influence of the strength of the field gradient on the D measurement? Does this mean that for glycerol content values above 40%, or crowding agents above 10-15% D values are overestimated because the field gradient is not strong enough? Please give details.

Page 4 Figure 1. It is awkward to draw the calculation with symbols and a solid line, which is, as I understand it, a visual guide. The dotted line on the experimental points is also a guide for the eye. To be consistent with the other figures (Figure 5) where the calculated D is in solid line, the authors should use solid line for calculations and symbols for experimental results. This remark is also true for figure 2 and 4 as well. It would also be preferable to express glycerol content in concentration (molar) rather than % by weight, which would enable comparisons with other gelling molecules, and with the tracer on a lattice model (figure 5) where concentration of PEG is used.

Page 5, figures 2 and 4. These two figures should be combined and, once again, the calculated value of D should only be represented by a solid line. As mentioned above, it would also be preferable to express PEG content in concentration.

Figure 5. Please express in concentration not in volume fraction.

Page 6 Figure 5. The alpha parameter is far from 1-2/pi = 0.36. Has it been adjusted by the fitting procedure? Please explain. This means that the network is not a square lattice. What would be the symmetry of the lattice for values of 0.1 and 0.075?

Page 7, Figure 6. The figure is not to scale and is therefore not correct! Please use values of 0.7, 2.5 and 3.9 nm hydrodynamic radius to represent PEG 300, 6000 and 40000. The protein has a Rh of 3.2 nm, so the right-hand side of the figure is clearly wrong because the protein and PEG 40000 are about the same size. The left-hand side is also wrong, as PEG 300 is much smaller than the protein. In addition, circles should be preferred as the only information on size is the hydrodynamic radius.

Page 7, figure 7. To my viewpoint this figure is not necessary as Figure 8 contains the same information.

Page 8, Figure 8. Same comments as for Figure 1.

Page 8, Figure 9. Same comments as for Figure 5.

Page 8, discussion on BSA and alpha-Casein. Diffusion coefficients (absolute values) are very different: they start at 8 and 5 in 10-11 m2/s respectively, and end at around 2-1 at high PEG concentration. Damping is therefore greater for BSA than for alpha-casein. Do you have any comments on this? Is it possible for alpha-casein to change conformation in a crowded environment? This question is related to the discussion of foldable and non-foldable IDPs. A further experiment using circular dichroism, or other techniques, would help to justify the statement in lines 276-277 that alpha-casein is one of the non-foldable IDPs.

Page 11, lines 360-361. Based on the experiments it is not reasonable to state that alpha-casein belongs to the non-foldable IDP family. A measure using a structural technique (CD, IR, NMR) is needed to conclude.

Written English is correct

Author Response

Dear Reviewer,

Sincerely,

Yuriy Zuev
